# Training deep learning based denoisers
# without ground truth data

**Shakarim Soltanayev**      **Se Young Chun**
Department of Electrical Engineering
Ulsan National Institute of Science and Technology (UNIST), Republic of Korea
`{shakarim,sychun}@unist.ac.kr`

## Abstract

Recently developed deep-learning-based denoisers often outperform state-of-the-art conventional denoisers, such as the BM3D. They are typically trained to minimize the mean squared error (MSE) between the output image of a deep neural network and a ground truth image. In deep learning based denoisers, it is important to use high quality noiseless ground truth data for high performance, but it is often challenging or even infeasible to obtain noiseless images in application areas such as hyperspectral remote sensing and medical imaging. In this article, we propose a method based on Stein's unbiased risk estimator (SURE) for training deep neural network denoisers only based on the use of noisy images. We demonstrate that our SURE-based method, without the use of ground truth data, is able to train deep neural network denoisers to yield performances close to those networks trained with ground truth, and to outperform the state-of-the-art denoiser BM3D. Further improvements were achieved when noisy test images were used for training of denoiser networks using our proposed SURE-based method. Code is available at https://github.com/Shakarim94/Net-SURE.

## 1 Introduction

Deep learning has been successful in various high-level computer vision tasks [1], such as image classification [2, 3], object detection [4, 5], and semantic segmentation [6, 7]. Deep learning has also been investigated for low-level computer vision tasks, such as image denoising [8, 9, 10, 11, 12], image inpainting [13], and image restoration [14, 15, 16]. In particular, image denoising is a fundamental computer vision task that yields images with reduced noise, and improves the execution of other tasks, such as image classification [8] and image restoration [16].

Deep learning based image denoisers [9, 11, 12] have yielded performances that are equivalent to or better than those of conventional state-of-the-art denoising techniques such as BM3D [17]. These deep denoisers typically train their networks by minimizing the mean-squared error (MSE) between the output of a network and the ground truth (noiseless) image. Thus, it is crucial to have high quality noiseless images for high performance deep learning denoisers. Thus far, deep neural network denoisers have been successful since high quality camera sensors and abundant light allow the acquisition of high quality, almost noiseless 2D images in daily environment tasks. Acquiring such high quality photographs is quite cheap with the use of smart phones and digital cameras.

However, it is challenging to apply currently developed deep learning based image denoisers with minimum MSE to some application areas, such as hyperspectral remote sensing and medical imaging, where the acquisition of noiseless ground truth data is expensive, or sometimes even infeasible. For example, hyperspectral imaging contains hundreds of spectral information per pixel, often leading to increased noise in hyperspectral imaging sensors [18]. Long acquisitions may improve image quality, but it is challenging to perform them with spaceborne or airborne hyperspectral imaging. Similarly,

in medical imaging, ultra high resolution 3D MRI (sub-millimeter resolution) often requires several hours of acquisition time for a single, high quality volume, but reducing acquisition time leads to increased noise. In X-ray CT, image noise can be substantially reduced by increasing the radiation dose. Recent studies on deep learning based image denoisers [19, 20] used CT images generated with normal doses as the ground truth so that denoising networks would be able to be trained to yield excellent performance. However, increased radiation dose leads to harmful effects in scanned subjects, while excessively high doses may saturate the CT detectors (e.g., in a similar manner to the acquisition of a photo of the sun without any filter). Thus, acquiring ground truth data with newly developed CT scanners seems challenging without compromising the subjects' safety.

Conventional denoising methods do not usually require noiseless ground truth images to perform denoising, but often require them for tuning parameters of image filters to elicit the best possible results (minimum MSE). In order to identify the optimal parameters of conventional denoisers without ground truth data, several works have been conducted [21, 22] using Stein's unbiased risk estimator (SURE) [23], which is an unbiased MSE estimator. For the popular non-local means (NLM) filter [24], the analytical form of SURE was used to optimize the denoiser performance [25, 26, 27]. For denoisers whose analytical forms of SURE are not available, Ramani *et al.* [28] proposed a Monte-Carlo-based SURE (MC-SURE) method to determine near-optimal denoising parameters based on the brute-force search of the parameter space. Deledalle *et al.* [29] investigated the approximation of a weak gradient of SURE to optimize parameters using the quasi-Newton algorithm. However, since this method requires the computation of the weak Jacobian, it is not applicable to high dimensional parameter spaces, such as deep neural networks.

We propose a SURE-based training method for deep neural network denoisers without ground truth data. In Section 2, we review key results elicited from SURE and MC-SURE. Subsequently, in Section 3, we describe our proposed method using MC-SURE and a stochastic gradient for training deep learning based image denoisers. In Section 4, simulation results are presented for (a) conventional state-of-the-art denoiser (BM3D), (b) deep learning based denoiser trained with BM3D as the ground truth, (c) the same deep neural network denoiser with the proposed SURE training without the ground truth, and (d) the same denoiser network with ground truth data as a reference. Section 5 concludes this article by discussing several potential issues for further studies.

## 2 Background

### 2.1 Stein's unbiased risk estimator

A signal (or image) with Gaussian noise can be modeled as,

$$\boldsymbol{y} = \boldsymbol{x} + \boldsymbol{n} \tag{1}$$

where $\boldsymbol{x} \in \mathbb{R}^K$ is an unknown signal in accordance with $\boldsymbol{x} \sim p(\boldsymbol{x})$, $\boldsymbol{y} \in \mathbb{R}^K$ is a known measurement, $\boldsymbol{n} \in \mathbb{R}^K$ is an *i.i.d.* Gaussian noise such that $\boldsymbol{n} \sim \mathcal{N}(\boldsymbol{0}, \sigma^2 \boldsymbol{I})$, and where $\boldsymbol{I}$ is an identity matrix. We denote $\boldsymbol{n} \sim \mathcal{N}(\boldsymbol{0}, \sigma^2 \boldsymbol{I})$ as $\boldsymbol{n} \sim \mathcal{N}_{0,\sigma^2}$. An estimator of $\boldsymbol{x}$ from $\boldsymbol{y}$ (or denoiser) can be defined as a function of $\boldsymbol{y}$ such that

$$\boldsymbol{h}(\boldsymbol{y}) = \boldsymbol{y} + \boldsymbol{g}(\boldsymbol{y}) \tag{2}$$

where $\boldsymbol{h}, \boldsymbol{g}$ are functions from $\mathbb{R}^K$ to $\mathbb{R}^K$. Accordingly, the SURE for $\boldsymbol{h}(\boldsymbol{y})$ can be derived as follows,

$$\eta(\boldsymbol{h}(\boldsymbol{y})) = \sigma^2 + \frac{\|\boldsymbol{g}(\boldsymbol{y})\|^2}{K} + \frac{2\sigma^2}{K} \sum_{i=1}^{K} \frac{\partial \boldsymbol{g}_i(\boldsymbol{y})}{\partial \boldsymbol{y}_i} = \frac{\|\boldsymbol{y} - \boldsymbol{h}(\boldsymbol{y})\|^2}{K} - \sigma^2 + \frac{2\sigma^2}{K} \sum_{i=1}^{K} \frac{\partial \boldsymbol{h}_i(\boldsymbol{y})}{\partial \boldsymbol{y}_i} \tag{3}$$

where $\eta : \mathbb{R}^K \to \mathbb{R}$ and $\boldsymbol{y}_i$ is the $i$th element of $\boldsymbol{y}$. For a fixed $\boldsymbol{x}$, the following theorem holds:

**Theorem 1** ([23, 30]). *The random variable $\eta(\boldsymbol{h}(\boldsymbol{y}))$ is an unbiased estimator of*

$$\mathrm{MSE}(\boldsymbol{h}(\boldsymbol{y})) = \frac{1}{K} \|\boldsymbol{x} - \boldsymbol{h}(\boldsymbol{y})\|^2$$

*or*

$$\mathbb{E}_{\boldsymbol{n} \sim \mathcal{N}_{0,\sigma^2}} \left\{ \frac{\|\boldsymbol{x} - \boldsymbol{h}(\boldsymbol{y})\|^2}{K} \right\} = \mathbb{E}_{\boldsymbol{n} \sim \mathcal{N}_{0,\sigma^2}} \left\{ \eta(\boldsymbol{h}(\boldsymbol{y})) \right\} \tag{4}$$

where $\mathbb{E}_{\boldsymbol{n} \sim \mathcal{N}_{0,\sigma^2}}\{\cdot\}$ is the expectation operator in terms of the random vector $\boldsymbol{n}$. Note that in Theorem 1, $\boldsymbol{x}$ is treated as a fixed, deterministic vector.

In practice, $\sigma^2$ can be estimated [28] and $\|\boldsymbol{y} - \boldsymbol{h}(\boldsymbol{y})\|^2$ only requires the output of the estimator (or denoiser). The last divergence term of (3) can be obtained analytically in some special cases, such as in linear or NLM filters [26]. However, it is challenging to calculate this term analytically for more general denoising methods.

## 2.2 Monte-Carlo Stein's unbiased risk estimator

Ramani *et al.* [28] introduced a fast Monte-Carlo approximation of the divergence term in (3) for general denoisers. For a fixed unknown true image $\boldsymbol{x}$, the following theorem is valid:

**Theorem 2** ([28]). *Let $\tilde{\boldsymbol{n}} \sim \mathcal{N}_{0,1} \in \mathbb{R}^K$ be independent of $\boldsymbol{n}$, $\boldsymbol{y}$. Then,*

$$\sum_{i=1}^{K} \frac{\partial \boldsymbol{h}_i(\boldsymbol{y})}{\partial \boldsymbol{y}_i} = \lim_{\epsilon \to 0} \mathbb{E}_{\tilde{\boldsymbol{n}}} \left\{ \tilde{\boldsymbol{n}}^{\mathrm{t}} \left( \frac{\boldsymbol{h}(\boldsymbol{y} + \epsilon\tilde{\boldsymbol{n}}) - \boldsymbol{h}(\boldsymbol{y})}{\epsilon} \right) \right\} \tag{5}$$

*provided that $\boldsymbol{h}(\boldsymbol{y})$ admits a well-defined second-order Taylor expansion. If not, this is still valid in the weak sense provided that $\boldsymbol{h}(\boldsymbol{y})$ is tempered.*

Based on Theorem 2, the divergence term in (3) can be approximated by one realization of $\tilde{\boldsymbol{n}} \sim \mathcal{N}_{0,1}$ and a fixed small positive value $\epsilon$:

$$\frac{1}{K} \sum_{i=1}^{K} \frac{\partial \boldsymbol{h}_i(\boldsymbol{y})}{\partial \boldsymbol{y}_i} \approx \frac{1}{\epsilon K} \tilde{\boldsymbol{n}}^{\mathrm{t}} \left( \boldsymbol{h}(\boldsymbol{y} + \epsilon\tilde{\boldsymbol{n}}) - \boldsymbol{h}(\boldsymbol{y}) \right) \tag{6}$$

where $\mathrm{t}$ is the transpose operator. This expression has been shown to yield accurate unbiased estimates of MSE for many conventional denoising methods $\boldsymbol{h}(\boldsymbol{y})$ [28].

# 3 Method

In this section, we will develop our proposed MC-SURE-based method for training deep learning based denoisers without noiseless ground truth images by assuming a Gaussian noise model in (1).

## 3.1 Training denoisers using the stochastic gradient method

A typical risk for image denoisers with the signal generation model (1) is

$$\mathbb{E}_{\boldsymbol{x} \sim p(\boldsymbol{x}), \boldsymbol{n} \sim \mathcal{N}_{0,\sigma^2}} \|\boldsymbol{x} - \boldsymbol{h}(\boldsymbol{y}; \boldsymbol{\theta})\|^2 \tag{7}$$

where $\boldsymbol{h}(\boldsymbol{y}; \boldsymbol{\theta})$ is a deep learning based denoiser parametrized with a large-scale vector $\boldsymbol{\theta}$. It is usually infeasible to calculate (7) exactly due to expectation operator. Thus, the empirical risk for (7) is used as a cost function as follows:

$$\frac{1}{N} \sum_{j=1}^{N} \|\boldsymbol{h}(\boldsymbol{y}^{(j)}; \boldsymbol{\theta}) - \boldsymbol{x}^{(j)}\|^2 \tag{8}$$

where $\{(\boldsymbol{x}^{(1)}, \boldsymbol{y}^{(1)}), \cdots, (\boldsymbol{x}^{(N)}, \boldsymbol{y}^{(N)})\}$ are the $N$ pairs of a training dataset, sampled from the joint distribution of $\boldsymbol{x}^{(j)} \sim p(\boldsymbol{x})$ and $\boldsymbol{n}^{(j)} \sim \mathcal{N}_{0,\sigma^2}$. Note that (8) is an unbiased estimator of (7).

To train the deep learning network $\boldsymbol{h}(\boldsymbol{y}; \boldsymbol{\theta})$ with respect to $\boldsymbol{\theta}$, a gradient-based optimization algorithm is used such as the stochastic gradient descent (SGD) [31], momentum, Nesterov momentum [32], or the Adam optimization algorithm [33]. For any gradient-based optimization method, it is essential to calculate the gradient of (7) with respect to $\boldsymbol{\theta}$ as follows,

$$\mathbb{E}_{\boldsymbol{x} \sim p(\boldsymbol{x}), \boldsymbol{n} \sim \mathcal{N}_{0,\sigma^2}} 2\nabla_{\boldsymbol{\theta}} \boldsymbol{h}(\boldsymbol{y}; \boldsymbol{\theta})^{\mathrm{t}} \left( \boldsymbol{h}(\boldsymbol{y}; \boldsymbol{\theta}) - \boldsymbol{x} \right). \tag{9}$$

Therefore, it is sufficient to calculate the gradient of the empirical risk (8) to approximate (9) for any gradient-based optimization.

In practice, calculating the gradient of (8) for large $N$ is inefficient since a small amount of well-shuffled training data can often approximate the gradient of (8) accurately. Thus, a mini-batch is

typically used for efficient deep neural network training by calculating the mini-batch empirical risk as follows:

$$\frac{1}{M}\sum_{j=1}^{M}\|\boldsymbol{h}(\boldsymbol{y}^{(j)};\boldsymbol{\theta})-\boldsymbol{x}^{(j)}\|^2 \tag{10}$$

where $M$ is the number of one mini-match. Equation (10) is still an unbiased estimator of (7) provided that the training data is randomly permuted every epoch.

## 3.2  Proposed training method for deep learning based denoisers

To incorporate MC-SURE into a stochastic gradient-based optimization algorithm for training, such as the SGD or the Adam optimization algorithms, we modify the risk (7) in accordance with

$$\mathbb{E}_{\boldsymbol{x}\sim p(\boldsymbol{x})}\left[\mathbb{E}_{\boldsymbol{n}\sim\mathcal{N}_{0,\sigma^2}}\left(\|\boldsymbol{x}-\boldsymbol{h}(\boldsymbol{y};\boldsymbol{\theta})\|^2|\boldsymbol{x}\right)\right]. \tag{11}$$

where (11) is equivalent to (7) owing to conditioning.

From Theorem 1, an unbiased estimator for $\mathbb{E}_{\boldsymbol{n}\sim\mathcal{N}_{0,\sigma^2}}\left(\|\boldsymbol{x}-\boldsymbol{h}(\boldsymbol{y};\boldsymbol{\theta})\|^2|\boldsymbol{x}\right)$ can be derived as

$$K\eta(\boldsymbol{h}(\boldsymbol{y};\boldsymbol{\theta})) \tag{12}$$

such that for a fixed $\boldsymbol{x}$,

$$\mathbb{E}_{\boldsymbol{n}\sim\mathcal{N}_{0,\sigma^2}}\left(\|\boldsymbol{x}-\boldsymbol{h}(\boldsymbol{y};\boldsymbol{\theta})\|^2|\boldsymbol{x}\right)=\mathbb{E}_{\boldsymbol{n}\sim\mathcal{N}_{0,\sigma^2}}\|\boldsymbol{x}-\boldsymbol{h}(\boldsymbol{y};\boldsymbol{\theta})\|^2=K\mathbb{E}_{\boldsymbol{n}\sim\mathcal{N}_{0,\sigma^2}}\eta(\boldsymbol{h}(\boldsymbol{y};\boldsymbol{\theta})).$$

Thus, using the empirical risk expression in (10), an unbiased estimator for (7) is

$$\frac{1}{M}\sum_{j=1}^{M}\left\{\|\boldsymbol{y}^{(j)}-\boldsymbol{h}(\boldsymbol{y}^{(j)};\boldsymbol{\theta})\|^2-K\sigma^2+2\sigma^2\sum_{i=1}^{K}\frac{\partial\boldsymbol{h}_i(\boldsymbol{y}^{(j)};\boldsymbol{\theta})}{\partial\boldsymbol{y}_i}\right\} \tag{13}$$

noting that no noiseless ground truth data $\boldsymbol{x}^{(j)}$ were used in (13).

Finally, the last divergence term in (13) can be approximated using MC-SURE so that the final unbiased risk estimator for (7) will be

$$\frac{1}{M}\sum_{j=1}^{M}\left\{\|\boldsymbol{y}^{(j)}-\boldsymbol{h}(\boldsymbol{y}^{(j)};\boldsymbol{\theta})\|^2-K\sigma^2+\frac{2\sigma^2}{\epsilon}(\tilde{\boldsymbol{n}}^{(j)})^{\mathrm{t}}\left(\boldsymbol{h}(\boldsymbol{y}^{(j)}+\epsilon\tilde{\boldsymbol{n}}^{(j)};\boldsymbol{\theta})-\boldsymbol{h}(\boldsymbol{y}^{(j)};\boldsymbol{\theta})\right)\right\} \tag{14}$$

where $\epsilon$ is a small fixed positive number and $\tilde{\boldsymbol{n}}^{(j)}$ is a single realization from the standard normal distribution for each training data $j$. In order to make sure that the estimator (14) is unbiased, the order of $\boldsymbol{y}^{(j)}$ should be randomly permuted and the new set of $\tilde{\boldsymbol{n}}^{(j)}$ should be generated at every epoch.

The deep learning based image denoiser with the cost function of (14) can be implemented using a deep learning development framework, such as TensorFlow [34], by properly defining the cost function. Thus, the gradient of (14) can be automatically calculated when the training is performed.

One of the potential advantages of our SURE-based training method is that we can use all the available data without noiseless ground truth images. In other words, we can train denoising, deep neural networks with the use of training and testing data. This advantage may further improve the performance of deep learning based denoisers.

Lastly, almost any deep neural network denoiser can utilize our MC-SURE-based training by modifying the cost function from (10) to (14) as far as it satisfies the condition in Theorem 2. Many deep learning based denoisers with differentiable activation functions (e.g., sigmoid) can comply with this condition. Some denoisers with piecewise differentiable activation functions (e.g., ReLU) can still utilize Theorem 2 in the weak sense since

$$\|\boldsymbol{h}(\boldsymbol{y};\boldsymbol{\theta})\|\leq C_0(1+\|\boldsymbol{y}\|^{n_0})$$

for some $n_0 > 1$ and $C_0 > 0$. Therefore, we expect that our proposed method should work for most deep learning image denoisers [8, 9, 10, 11, 12].

# 4  Simulation results

In this section, denoising simulation results are presented with the MNIST dataset using a simple stacked denoising autoencoder (SDA) [8], and a large-scale natural image dataset using a deep convolutional neural network (CNN) image denoiser (DnCNN) [11].

All of the networks presented in this section (denoted by NET, which can be either SDA or DnCNN) were trained using one of the following two optimization objectives: (MSE) the minimum MSE between a denoised image and its ground truth image in (10) and (SURE) our proposed minimum MC-SURE without ground truth in (14). NET-MSE methods generated noisy training images at every epochs in accordance with [11], while our proposed NET-SURE methods used only noisy images obtained before training. We also propose the SURE-T method which utilized noisy test images with noisy training images and without ground truth data. Table 1 summarizes all simulated configurations including conventional state-of-the-art image denoiser, BM3D [17], that did not require any training, or the use of any ground truth data. Code is available at https://github.com/Shakarim94/Net-SURE.

Table 1: Summary of simulated denoising methods. NET can be either SDA or DnCNN.

| Method | Description |
|---|---|
| BM3D | Conventional state-of-the-art method |
| NET-BM3D | Optimizing MSE with BM3D output as ground truth |
| NET-SURE | Optimizing SURE without ground truth |
| NET-SURE-T | Optimizing SURE without ground truth, but with noisy test data |
| NET-MSE-GT | Optimizing MSE with ground truth |

## 4.1  Results: MNIST dataset

We performed denoising simulations with the MNIST dataset. Noisy images were generated based on model (1) with two noise levels (one with $\sigma = 25$ and the other with $\sigma = 50$). For the experiments on the MNIST dataset which comprised $28 \times 28$ pixels, a simple SDA network was chosen [8]. Decoder and encoder networks each consisted of two convolutional layers (kernel size $3 \times 3$) with sigmoid activation functions, each of which had a stride of two (both conv and conv transposed). Thus, a training sample with a size of $28 \times 28$ is downsampled to $7 \times 7$, and then upsampled to $28 \times 28$.

SDA was trained to output a denoised image using a set of 55,000 training and 5,000 validation images. The performance of the model was tested with 100 images chosen randomly from the default test set of 10,000 images. For all cases, SDA was trained with the Adam optimization algorithm [33] with the learning rate of 0.001 for 100 epochs. The batch size was set to 200 (bigger batch sizes did not improve the performance). The $\epsilon$ value in (6) was set to 0.0001.

Our proposed methods SDA-SURE, SDA-SURE-T yielded a comparable performance to SDA-MSE-GT (only 0.01-0.04 dB difference) and outperformed the conventional BM3D for all simulated noise levels, $\sigma = 25, 50$, as shown in Table 2.

Table 2: Results of denoisers for MNIST (performance in dB). Means of 10 experiments are reported.

| Methods | BM3D | SDA-REG | SDA-SURE | SDA-SURE-T | SDA-MSE-GT |
|---|---|---|---|---|---|
| $\sigma = 25$ | 27.53 | 25.07 | **28.35** | **28.39** | 28.35 |
| $\sigma = 50$ | 21.82 | 19.85 | **25.23** | **25.24** | 25.24 |

Figure 1 illustrates the visual quality of the outputs of the simulated denoising methods at high noise levels ($\sigma = 50$). All SDA-based methods clearly outperform the conventional BM3D method based on visual inspection (BM3D image looks blurry compared to other SDA-based results), while it is indistinguishable for the simulation results among all SDA methods with different cost functions and training sets. These observations were confirmed by the quantitative results shown in Table 2. All SDA-based methods outperformed BM3D significantly, but there were very small differences among all the SDA methods, even when noisy test data were used.

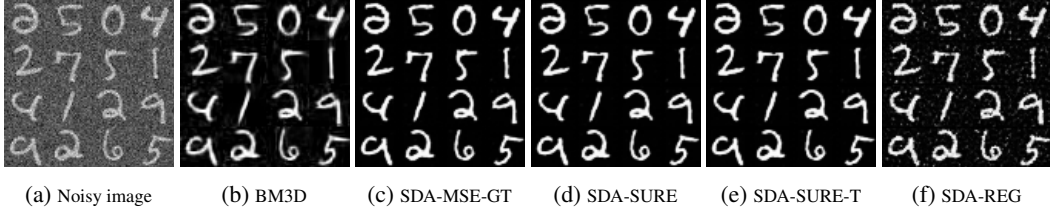

| (a) Noisy image | (b) BM3D | (c) SDA-MSE-GT | (d) SDA-SURE | (e) SDA-SURE-T | (f) SDA-REG |

Figure 1: Denoising results of SDA with various methods for MNIST dataset at a noise level of $\sigma$=50.

## 4.2 Regularization effect of deep neural network denoisers

Parametrization of deep neural networks with different number of parameters and structures may introduce a regularization effect in training denoisers. We further investigated this regularization effect by training the SDA to minimize the MSE between the output of SDA and the input noisy image (SDA-REG). In the case of a noise level of $\sigma = 50$, early stopping rule was applied when the network started to overfit the noisy dataset after the first few epochs. The performance of this method was significantly worse than those of all other methods with PSNR values of 25.07 dB ($\sigma = 25$) and 19.85 dB ($\sigma = 50$), as shown in Table 2. These values are approximately 2 dB lower than the PSNRs of BM3D. Noise patterns are visible, as shown in Figure 1. This shows that the good performance of SDA is not attributed to its structure only, but also depends on the optimization of MSE or SURE.

## 4.3 Accuracy of MC-SURE approximation

A small value must be assigned to $\epsilon$ in (6) for accurate estimation of SURE. Ramani *et al.* [28] have observed that $\epsilon$ can take a wide range of values and its choice is not critical. According to our preliminary experiments for the SDA with an MNIST dataset, any choice for $\epsilon$ in the range of $[10^{-2}, 10^{-7}]$ worked well so that the SURE approximation matches close to the MSE during training, as illustrated in Figure 2 (middle). Extremely small values $\epsilon < 10^{-8}$ resulted in numerical instabilities, as shown in Figure 2 (right). On the contrary, when $\epsilon > 10^{-1}$, the approximation in (6) becomes substantially inaccurate. Figure 3 illustrates how the performance of SDA-SURE is affected by the $\epsilon$ value. However, note that these values are only for SDA trained with the MNIST dataset. The admissible range of $\epsilon$ depends on $\boldsymbol{h}_i(\boldsymbol{y}; \boldsymbol{\theta})$. For example, we observed that a suitable $\epsilon$ value must be carefully selected in other cases, such as DnCNN with, large-scale parameters and high resolution images for improved performance.

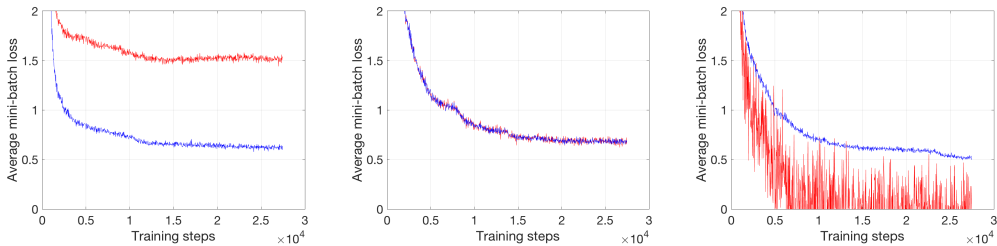

Figure 2: Loss curves for the training of SDA with MSE (blue) and its corresponding MC-SURE (red) using different $\epsilon$ values, $\epsilon = 1$ (left), $\epsilon = 10^{-5}$ (middle), and $\epsilon = 10^{-9}$ (right). MC-SURE accurately approximates the true MSE for a wide range of $\epsilon$.

The accuracy of MC-SURE also depends on the noise level $\sigma$. It was observed that the SURE loss curves become noisier compared to MSE loss curves as $\sigma$ increases. However, they still followed similar trends and yielded similar average PSNRs on MNIST dataset as shown in Figure 4. We observed that after $\sigma = 350$, SURE loss curves started to become too noisy and thus deviated from the trends of their corresponding MSE loss curves. Conversely, noise levels $\sigma > 300$ were too high for both SDA-based denoisers and BM3D, so that they were not able to output recognizable digits. Therefore, SDA-SURE can be trained effectively on adequately high noise levels so that it can yield a performance that is comparable to SDA-MSE-GT and can consistently outperform BM3D.

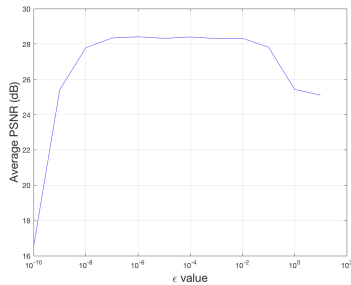
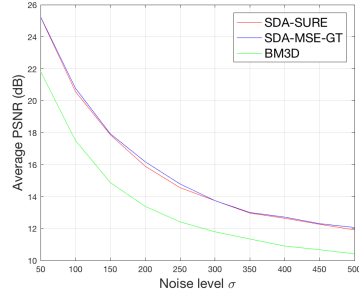

Figure 3: Performance of SDA-SURE for different $\epsilon$ values at $\sigma = 25$.

Figure 4: Performance of denoising methods at different $\sigma$ values.

## 4.4 Results: high resolution natural images dataset

To demonstrate the capabilities of our SURE-based deep learning denoisers, we investigated a deeper and more powerful denoising network called DnCNN [11] using high resolution images. DnCNN consisted of 17 layers of CNN with batch normalization and ReLU activation functions. Each convolutional layer had 64 filters with sizes of $3 \times 3$. Similar to [11], the network was trained with 400 images with matrix sizes of $180 \times 180$ pixels. In total, $1772 \times 128$ image patches with sizes of $40 \times 40$ pixels were extracted randomly from these images. Two test sets were used to evaluate performance: one set consisted of 12 widely used images (Set12) [17], and the other was a BSD68 dataset. For DnCNN-SURE-T, additional $808 \times 128$ image patches were extracted from these noisy test images, and were then added to the training dataset. For all cases, the network was trained with 50 epochs using the Adam optimization algorithm with an initial learning rate of 0.001, which eventually decayed to 0.0001 after 40 epochs. The batch size was set to 128 (note that bigger batch sizes did not improve performance). Images were corrupted at three noise levels ($\sigma = 25, 50, 75$).

DnCNN used residual learning [11] whereby the network was forced to learn the difference between noisy and ground truth images. The output residual image was then subtracted from the input noisy image to yield the estimated image. In other words, our network was trained with SURE as

$$h(y; \theta) = y - \mathbf{CNN}_{\theta}(y) \tag{15}$$

where $\mathbf{CNN}_{\theta}(.)$ is the DnCNN that is being trained using residual learning. For DnCNN, selecting an appropriate $\epsilon$ value in (6) turned out to be important for a good denoising performance. To achieve stable training with good performance, $\epsilon$ had to be tuned for each of the chosen noise levels of $\sigma = 25, 50, 75$. We observed that the optimal value for $\epsilon$ was proportional to $\sigma$ as shown in [29]. All the experiments were performed with the setting of $\epsilon = \sigma \times 1.4 \times 10^{-4}$.

With the use of an NVidia Titan X GPU, the training process took approximately 7 hours for DnCNN-MSE-GT and approximately 11 hours for DnCNN-SURE. SURE based methods took more training

Table 3: Results of denoising methods on 12 widely used images (Set12) (performance in dB).

| IMAGE | C. MAN | HOUSE | PEPPERS | STARFISH | MONARCH | AIRPLANE | PARROT | LENA | BARBARA | BOAT | MAN | COUPLE | Average |
|---|---|---|---|---|---|---|---|---|---|---|---|---|---|
| | | | | | $\sigma = 25$ | | | | | | | | |
| BM3D | 29.47 | **33.00** | 30.23 | 28.58 | 29.35 | 28.37 | 28.89 | **32.06** | **30.64** | 29.78 | 29.60 | 29.70 | 29.97 |
| DnCNN-BM3D | 29.34 | 31.99 | 30.13 | 28.38 | 29.21 | 28.46 | 28.91 | 31.53 | 28.89 | 29.6 | 29.52 | 29.54 | 29.63 |
| DnCNN-SURE | **29.80** | 32.70 | **30.58** | 29.08 | 30.11 | 28.94 | 29.17 | 32.06 | 29.16 | 29.84 | 29.89 | 29.76 | 30.09 |
| DnCNN-SURE-T | **29.86** | 32.73 | 30.57 | **29.11** | 30.13 | 28.93 | **29.26** | 32.08 | 29.44 | **29.86** | **29.91** | **29.78** | **30.14** |
| DnCNN-MSE-GT | 30.14 | 33.16 | 30.84 | 29.4 | 30.45 | 29.11 | 29.36 | 32.44 | 29.91 | 30.11 | 30.08 | 30.06 | 30.42 |
| | | | | | $\sigma = 50$ | | | | | | | | |
| BM3D | 26.00 | **29.51** | 26.58 | 25.01 | 25.78 | 25.15 | 25.98 | **28.93** | **27.19** | 26.62 | 26.79 | 26.46 | **26.67** |
| DnCNN-BM3D | 25.76 | 28.43 | 26.5 | 24.9 | 25.66 | 25.15 | 25.82 | 28.36 | 25.3 | 26.5 | 26.6 | 26.17 | 26.26 |
| DnCNN-SURE | **26.48** | 29.14 | **26.77** | 25.38 | 26.50 | 25.66 | 26.21 | 28.79 | 24.86 | 26.78 | 26.97 | 26.51 | **26.67** |
| DnCNN-SURE-T | 26.47 | 29.20 | **26.78** | 25.39 | 26.53 | 25.65 | 26.21 | 28.81 | 25.23 | **26.79** | 26.97 | 26.48 | **26.71** |
| DnCNN-MSE-GT | 27.03 | 29.92 | 27.27 | 25.65 | 26.95 | 25.93 | 26.43 | 29.31 | 26.17 | 27.12 | 27.22 | 26.94 | 27.16 |
| | | | | | $\sigma = 75$ | | | | | | | | |
| BM3D | 24.58 | **27.45** | **24.69** | 23.19 | 23.81 | 23.38 | **24.22** | **27.14** | **25.08** | 25.05 | 25.30 | **24.73** | **24.89** |
| DnCNN-BM3D | 24.11 | 27.02 | 24.48 | 23.09 | 23.73 | 23.40 | 24.06 | 27.11 | 23.80 | 24.84 | 25.19 | 24.59 | 24.62 |
| DnCNN-SURE | **24.65** | 27.16 | 24.49 | 23.25 | 24.10 | 23.52 | 24.13 | 26.92 | 23.02 | 25.09 | 25.37 | 24.70 | 24.70 |
| DnCNN-SURE-T | **24.82** | 27.34 | 24.58 | **23.34** | 24.25 | 23.56 | 24.44 | 27.03 | 23.07 | **25.17** | **25.45** | 24.78 | 24.82 |
| DnCNN-MSE-GT | 25.46 | 28.04 | 25.22 | 23.62 | 24.81 | 23.97 | 24.71 | 27.60 | 23.88 | 25.53 | 25.68 | 25.13 | 25.30 |

Table 4: Results of denoising methods on BSD68 dataset (performance in dB).

| Methods | BM3D | DnCNN-BM3D | DnCNN-SURE | DnCNN-SURE-T | DnCNN-MSE-GT |
|---|---|---|---|---|---|
| $\sigma = 25$ | 28.56 | 28.54 | **28.97** | **29.00** | 29.20 |
| $\sigma = 50$ | 25.62 | 25.44 | **25.93** | **25.95** | 26.22 |
| $\sigma = 75$ | 24.20 | 24.09 | **24.31** | **24.37** | 24.66 |

time than MSE based methods because of the additional divergence calculations executed to optimize the MC-SURE cost function. For the DnCNN-SURE-T method, it took approximately 15 hours to complete the training owing to the larger dataset.

Tables 3 and 4 present denoising performance data using (a) the BM3D denoiser [17], (b) a state-of-the-art deep CNN (DnCNN) image denoiser trained with MSE [11], and (c) the same DnCNN image denoiser trained with SURE without the use of noiseless ground truth images, for different dataset variations (as shown in Table 1). The MSE-based DnCNN image denoiser with ground truth data, DnCNN-MSE-GT, yielded the best denoising performance compared to other methods, such as the BM3D, which is consistent with the results in [11].

As seen in Table 3, for the Set12 dataset, SURE-based denoisers achieved performances comparable to or better than that for BM3D for noise levels $\sigma = 25$ and 50. In contrast, for higher noise levels ($\sigma = 75$), DnCNN-SURE and DnCNN-SURE-T yielded lower average PSNR values by 0.19 dB and 0.07 dB than BM3D. DnCNN-SURE-T outperformed DnCNN-SURE in all cases, and had considerably better performance on some images, such as "Barbara." BM3D had exceptionally good denoising performance on the "Barbara" image (up to 2.33 dB better PSNR), and even outperformed the DnCNN-MSE-GT method.

In the case of the BSD68 dataset in Table 4, SURE-based methods outperformed BM3D for all the noise levels. Unlike the case of the Set12 images, we observed that DnCNN-SURE had a significantly better performance than BM3D, and yielded increased average PSNR values by 0.11 - 0.41 dB. It was also observed that DnCNN-SURE-T benefited from the utilization of noisy test images and improved the average PSNR of DnCNN-SURE.

Differences among the performances of denoisers in Tables 3 and 4 can be explained by the working principle of BM3D. Since BM3D looks for similar image patches for denoising, repeated patterns (as in the "Barbara" image) and flat areas (as in "House" image) can be key factors to generating improved denoising results. One of the advantages of DnCNN-SURE over BM3D is that it does not suffer from rare patch effects. If the test image is relatively detailed and does not contain many similar patterns, BM3D will have poorer performance than the proposed DnCNN-SURE method. Note that the DnCNN-BM3D method that trains networks by optimizing MSE with BM3D denoised images as the ground truth yielded slightly worse performance than the BM3D itself (Tables 3, 4).

Figure 5 illustrates the denoised results for an image from the BSD68 dataset. Visual quality assessment indicated that BM3D yielded blurrier images and thus yielded worse PSNR compared to the results generated by deep neural network denoisers. DnCNN-MSE-GT had the best denoised image with the highest PSNR of 26.85 dB, while both SURE methods yielded very similar performances in accordance with PSNR and visual quality assessment.

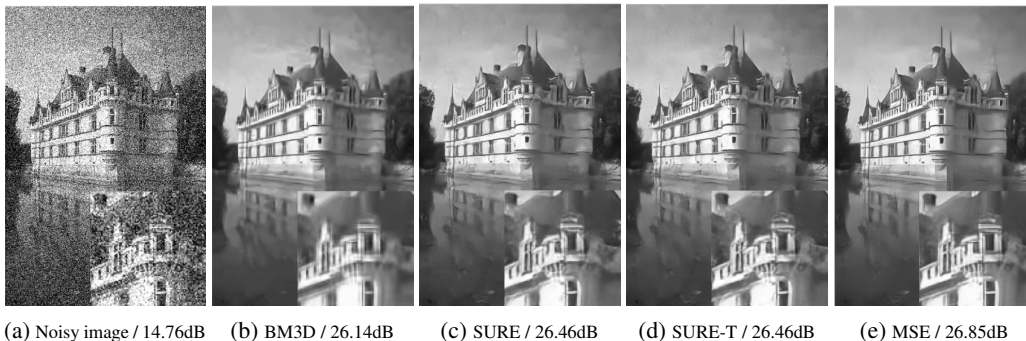

(a) Noisy image / 14.76dB    (b) BM3D / 26.14dB    (c) SURE / 26.46dB    (d) SURE-T / 26.46dB    (e) MSE / 26.85dB

Figure 5: Denoising results of an image from the BSD68 dataset for $\sigma$=50

# 5 Discussion

It has been shown that a single deep denoiser network, such as DnCNN, can be trained to deal with multiple noise levels (e.g. $\sigma = [0, 55]$) [11]. Thus, our work can be easily generalized to train denoisers for multiple noise levels by modifying (14) to have $\sigma^{(j)}$ and $\epsilon^{(j)}$ for the SURE risk of the $j$th image patch. For example, $\epsilon$ values for image patches can be specified by our formula $\epsilon^{(j)} = \sigma^{(j)} \times 1.4 \times 10^{-4}$ for DnCNN. In fact, SURE based DnCNN networks were trained to handle a wide range of noise levels (blind denoising) in [35].

SURE-based methods used noisy training images only, but SURE-T methods used both noisy training and test images. SURE-T methods yielded a slightly better denoising performance than SURE-based methods (approximately 0.02 - 0.06 dB and 0.04 - 0.12 dB for BSD68 and Set12 datasets, respectively) with considerably increased the overall inference time. Thus, at this moment, SURE-T methods do not seem to have considerable benefits compared to SURE-based methods. However, developing a hybrid method that first trains networks using SURE with training data, and then fine-tunes the network using SURE with testing data, could be potentially useful to reduce the overall inference time. Finding a connection between SURE-T and "deep image prior" that used noisy test images for denoising [36] can constitute interesting future work.

Our proposed SURE-based deep learning denoiser can be useful for applications with considerably large amounts of noisy images, but with few noiseless images, or with expensive noiseless images. Deep learning based denoising research is still evolving, and it may be even possible for our SURE-based training method to achieve significantly better performances than BM3D, or other conventional state-of-the-art denoisers, when it is applied to novel deep neural network denoisers. Further investigation will be needed for high performance denoising networks for synthetic and real noise.

In this work, Gaussian noise with known variance was assumed in all simulations. However, there are several noise estimation methods that can be used with SURE (see [28] for details). SURE can incorporate a variety of noise distributions other than Gaussian noise. For example, SURE has been used for parameter selection of conventional filters for a Poisson distribution [29]. Generalized SURE for exponential families has been proposed [37] so that other common noise types in imaging systems can be potentially considered for SURE-based methods. It should be noted that SURE does not require any prior knowledge on images. Thus, potentially it can be applied to the measurement domain for different applications, such as medical imaging. Owing to noise correlation (colored noise) in the image domain (e.g., based on the Radon transform in the case of CT or PET), further investigations will be necessary to apply our proposed method directly to the image domain.

Note that unlike (7), the existence of the minimizer for (14) should be considered with care since it is theoretically possible that (14) becomes negative infinity due to the divergence term in (14). However, in practice, this issue can be easily addressed by introducing a regularizer (weight decay), with a deep neural network structure so that denoisers can impose regularity conditions on function $h$ (e.g., bounded norm of $\nabla h$), either by choosing an adequate $\epsilon$ value, or by using proper training data. Lastly, note that we derived (14), an unbiased estimator for MSE, assuming a fixed $\boldsymbol{\theta}$. Thus, there is no guarantee that the resulting estimator (denoiser) that is tuned by SURE will be unbiased [38].

# 6 Conclusion

We proposed a MC-SURE based training method for general deep learning denoisers. Our proposed method trained deep neural network denoisers without noiseless ground truth data so that they could yield comparable denoising performances to those elicited by the same denoisers that were trained with noiseless ground truth data, and outperform the conventional state-of-the-art BM3D. Our SURE-based training method worked successfully in the simple SDA [8], and in the case of the state-of-the-art DnCNN [11] without the use of ground truth images.

### Acknowledgments

This work was supported partly by Basic Science Research Program through the National Research Foundation of Korea(NRF) funded by the Ministry of Education(NRF-2017R1D1A1B05035810) and partly by the Technology Innovation Program or Industrial Strategic Technology Development Program (10077533, Development of robotic manipulation algorithm for grasping/assembling with

the machine learning using visual and tactile sensing information) funded by the Ministry of Trade, Industry & Energy (MOTIE, Korea).

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
