[Supplementary Material]

# Supplementary material: Training deep learning based denoisers without ground truth data

**Shakarim Soltanayev**     **Se Young Chun**
Department of Electrical Engineering
Ulsan National Institute of Science and Technology (UNIST)
Ulsan, Republic of Korea
{shakarim,sychun}@unist.ac.kr

Table S1: Results of denoising methods on both BSD68 and Set12 datasets combined (performance in dB). For the DnCNN-BM3D method, the network was trained by optimizing the MSE between BM3D-denoised images and the output images of DnCNN. This method achieved the worst performance among all denoisers.

| Methods | BM3D | DnCNN-BM3D | DnCNN-SURE | DnCNN-SURE-T | DnCNN-MSE-GT |
|---|---|---|---|---|---|
| $\sigma = 25$ | 28.77 | 28.70 | **29.14** | **29.17** | 29.38 |
| $\sigma = 50$ | 25.78 | 25.56 | **26.04** | **26.06** | 26.35 |
| $\sigma = 75$ | 24.30 | 24.17 | **24.37** | **24.44** | 24.76 |

Table S2: Results of denoising methods at low noise levels on the Set12 dataset (performance in dB).

| Methods | BM3D | DnCNN-SURE | DnCNN-MSE-GT |
|---|---|---|---|
| $\sigma = 5$ | 38.03 | **38.16** | 38.23 |
| $\sigma = 10$ | 34.37 | **34.58** | 34.72 |
| $\sigma = 15$ | **32.36** | **32.36** | 32.78 |

Table S3: Results of denoising methods at low noise levels on the BSD68 dataset (performance in dB).

| Methods | BM3D | DnCNN-SURE | DnCNN-MSE-GT |
|---|---|---|---|
| $\sigma = 5$ | 37.56 | **37.81** | 37.87 |
| $\sigma = 10$ | 33.38 | **33.72** | 33.82 |
| $\sigma = 15$ | 31.07 | **31.34** | 31.66 |

(a) Noisy image / 20.19dB

(b) BM3D / 29.16dB

(c) DnCNN-BM3D / 29.23dB

(d) DnCNN-SURE / 29.68dB

(e) DnCNN-SURE-T / 29.69dB

(f) DnCNN-MSE-G / 29.81dB

Figure S1: Denoising results of an image from the BSD68 dataset at $\sigma$=25. Deep learning based methods yielded sharper images compared to BM3D.

(a) Noisy image / 14.76dB

(b) BM3D / 27.19dB

(c) DnCNN-BM3D / 25.30dB

(d) DnCNN-SURE / 24.86dB

(e) DnCNN-SURE-T / 25.23dB

(f) DnCNN-MSE-G / 26.17dB

Figure S2: Denoising results of "Barbara" image at $\sigma$=50. BM3D yielded exceptionally good performance on this image owing to many repeated patterns. It even outperformed DnCNN-MSE-G for this special case.

(a) Noisy image / 14.15dB

(b) BM3D / 25.78dB

(c) DnCNN-BM3D / 25.66dB

(d) DnCNN-SURE / 26.50dB

(e) DnCNN-SURE-T / 26.53dB

(f) DnCNN-MSE-G / 26.95dB

Figure S3: Denoising results of "Monarch" image at $\sigma$=50. Deep learning based methods yielded sharper images compared to the BM3D.

(a) Noisy image / 11.77dB     (b) BM3D / 26.21dB     (c) DnCNN-BM3D / 25.83dB

(d) DnCNN-SURE / 26.64dB     (e) DnCNN-SURE-T / 26.67dB     (f) DnCNN-MSE-G / 27.10dB

Figure S4: Denoising results of an image from the BSD68 dataset at $\sigma$=75. Deep learning based methods yielded sharper images compared to the BM3D.