[Reviews · NeurIPS 2018]

Reviewer 1



The submission "Training deep learning denoisers without ground truth data" transfers methods from risk estimation to the setting of convolutional neural networks for denoising. The usual minimization of the l2-loss between ground truth and training data is replaced by the minimization over an unbiased estimator over training data, sampled in a Monte-Carlo fashion. The submission highlights how previous techniques for unbiased parameter estimation can be translated into the CNNs and shows very intriguing results, training without ground truth data. A missing aspect that has to be addressed is the existence of minimizer of the SURE estimator (equation (13)) - it is easy to contruct simple (e.g. linear) networks and corresponding training data such that no minimizer \theta of (13) exists. The function value is not necessarily bounded from below, and the infimum over (13) becomes minus infinity. How can such cases be excluded, either by assumptions on the data / the number of free parameters, or by additional regularization on theta? As minor comments - I think it would be fair to state that the interesting idea of including test data during training also increases the inference speed significantly (e.g. to the 15 hours mentioned). - Along the lines of including the test data during training, the authors could train on as few as a single test image, yielding an interesting connection to the recent work on "Deep Image Priors" by Ulyanov et al. - Possibly related prior work on using the SURE estimator for denoising can be found, e.g., in Donoho and Johnstone, "Adapting to unknown smoothness via Wavelet Shrinkage" or Zhang and Desai, "Adaptive Denoising based on SURE risk". - It should be noted, that although equation (13) is an unbiased risk estimator for the MSE loss for fixed parameters theta, the minimimum over theta is not necessarily an unbiased estimation of the MSE loss of the estimator parametrized by this minimizer, see, e.g., section 1.3 in Tibsharani and Rosset's "Excess Optimism: How Biased is the Apparent Error of an Estimator Tuned by SURE?". - Please have your submission proof-read for English style and grammar issues with a particular focus on missing articles. - I am wondering if completing the square in equation (14), ariving at a term \|y^{(j)} + \frac{\sigma^2}{\epsilon} \tilde{n}^{(j)} - h(y^{(j)}; \theta)\|^2 and one remaining linear term provides an intering point of view. What happens if one just trains with the above quadratic term? This is just a question out of curiosity and does not have to be addressed. In summary, the work is interesting and the ideas are clearly explained. The experiments are detailed and well-formulated and show promising results. The paper is, in general, well structured and easy to read. I recommend the acceptance of this manuscript. I appreciate the clarifications of the authors in their rebuttal. Although it is a minor technical issue (typically avoided e.g. by weight decay), I'd like to point out that the existence of minimizers is not only a question of the network architecture, but also of the training data. I agree with the other reviewers that the limitation to (fixed) Gaussian noise is severe in practice, but I'm willing to accept such a limitation as this is one of the very first papers to train denoising networks without any ground truth. Because the general idea of the paper is very interesting and the first results are promising, I keep my score and recommend the acceptance of this manuscript.

Reviewer 2



This submission deals with image denoising using deep learning when no noiseless ground-truth information is available. To this end, an unbiased estimate of the MSE cost based on SURE estimator is adopted, where the divergence terms are estimated using the Monte-Carlo approximation. Experiments are performed using synthetic data with Gaussian noise added to the real natural images. The observations indicate close performance to the MSE training with noiseless ground-truth data. This is an important problem. The proposed solution for the Gaussian noise is also reasonable given the literature on SURE estimator. The Gaussian noise assumption however limits the scope and applicability of this method to deal with more complicated real-world noise structures. More comments: 1) It is not clear why the gradient is approximated in eqn. (5) since the end-to-end gradient can be easily calculated using optimizers such as TensorFlow? 2) This model is limited to Gaussian noise, which is not practical. How generalizable could it be to other noise structures that are more common in practice? The authors argue in the conclusions section about it, but more extensive experiments are needed to verify that for a real-world noisy dataset. 3) What is the breaking point for the performance of MC-SURE in terms of the noise variance? It would be useful to plot the test PSNR versus the noise level to see that. 4) Would the network be trained seperately for each noise level? How can one generalize that to a model that can deal with all noise levels since in practice the noise level may not be identical for all train and test images. Reviewer's opinion about the authors rebuttal The rebuttal does not address my main concerns. I however appreciate the authors clarifying how the model can deal with multiple noise levels. The authors argument about the gradient approximation. Why can’t one calculate the entire Jacobian easily with chain rule? In addition, I don’t agree with the argument about the Gaussian noise. The authors indicate that extension to other simple noise structures such as Poisson is non-trivial, which I agree with the authors. It seems extension to colored noise distributions that is more practical is even harder. Overall, I am not satisfied with the authors response, but I believe this submission is interesting and has merits, but needs to be improved and addresses the practical concerns.

Reviewer 3



This paper proposed to train deep networks for image denoising based on Monte-Carlo Stein's unbiased risk estimator (MC-SURE). The advantage is that the proposed MC-SURE based method enables training deep denoising networks without noiseless ground truth data. Experiments show that the proposed MC-SURE based method achieves comparable denoising performance for Gaussian noise to the same denoisers that are trained with noiseless ground truth data. Although the idea is very attractive, there are some problems to be clarified in the paper. 1) A curly brace is missing in the equation (14). 2) There is an error in Fig. 1, which is not clear and confusing. 3) The proposed method only works for Gaussian noise case with known standard variance, which is a very strong limitation. How about other types of noise? To make the whole method more practical, how about dealing with the noisy image data with unknown standard variance? 4) From Eq. (14), it is obvious that the proposed MC-SURE based method is dependent on the standard variance of Gaussian noise, i.e. sigma. In the experiments, each sigma needs training one network. Is it possible to train one network for several noise levels based MC-SURE? 5) For natural images, the author just sets epsilon to be a fixed. I suggest plotting the sensitivity figure of epsilon for different noise levels, so that the authors can understand the effect of epsilon. 6) In addition to 25, 50, 75, the results of other noise levels should be provided, such as 5, 10, 15.